# Finding Needles in a Haystack: Using Geo-References to Enhance the Selection and Utilization of Landraces in Breeding for Climate-Resilient Cultivars of Upland Cotton (*Gossypium hirsutum* L.)

**DOI:** 10.3390/plants10071300

**Published:** 2021-06-26

**Authors:** Junghyun Shim, Nonoy B. Bandillo, Rosalyn B. Angeles-Shim

**Affiliations:** 1Department of Plant and Soil Science, College of Agricultural Sciences and Natural Resources, Texas Tech University, Lubbock, TX 79409-2122, USA; junghyun.shim@ttu.edu; 2Department of Plant Sciences, North Dakota State University, Fargo, ND 58108-6050, USA; nonoy.bandillo@ndsu.edu

**Keywords:** landraces, core collections, focused identification of germplasm strategy, environmental association analysis, landscape genomics

## Abstract

The genetic uniformity of cultivated cotton as a consequence of domestication and modern breeding makes it extremely vulnerable to abiotic challenges brought about by major climate shifts. To sustain productivity amidst worsening agro-environments, future breeding objectives need to seriously consider introducing new genetic variation from diverse resources into the current germplasm base of cotton. Landraces are genetically heterogeneous, population complexes that have been primarily selected for their adaptability to specific localized or regional environments. This makes them an invaluable genetic resource of novel allelic diversity that can be exploited to enhance the resilience of crops to marginal environments. The utilization of cotton landraces in breeding programs are constrained by the phenology of the plant and the lack of phenotypic information that can facilitate efficient selection of potential donor parents for breeding. In this review, the genetic value of cotton landraces and the major challenges in their utilization in breeding are discussed. Two strategies namely Focused Identification of Germplasm Strategy and Environmental Association Analysis that have been developed to effectively screen large germplasm collections for accessions with adaptive traits using geo-reference-based, mathematical modelling are highlighted. The potential applications of both approaches in mining available cotton landrace collections are also presented.

## 1. Introduction

The sessile nature of plants predisposes them to a constant array of environmental challenges such as drought, flooding, temperature extremes, and soil salinity and toxicity. More often than not, plants are exposed not just to one but a combination of these stresses at any given time [1,2]. In the High Plains of West Texas for example, strong radiation tends to occur together with exceedingly high temperature, low air humidity and water deficit. Together, these not only limit the productivity of field crops but also potentially weaken defense response mechanisms, making plants more vulnerable to diseases and pests [1].

In 2007, the Food and Agriculture Organization of the United Nations reported that as much as 96.5% of the rural land area worldwide are already under intense pressure from temperature extremes and poor soils [3]. Over time, abiotic stresses are expected to put more pressure on agricultural production as global temperature continues to rise and push agro-environments into a steady decline. This is in addition to the biotic pressures from the expansion of pest populations to new territories and the emergence of new strains of pathogens due to major shifts in climate [4].

Like most field crops, the growth, development and productivity of cotton are largely determined by the prevailing environmental conditions within a cropping season. Its overall response to abiotic challenges, however, is more complicated owing to its indeterminate growth habit and complex fruiting patterns [5]. In recent years, incidences of severe cotton damage due to worsening agro-environments have been increasing worldwide. In 2011 for example, historic drought combined with record high temperature and unprecedented low precipitation devastated cotton production in Texas, USA. Approximately 55% of cotton fields were abandoned that year, leading to subsequent financial losses amounting to 7.62 billion US dollars [6,7]. In Pakistan, the unexpected occurrence of heavy rains that coincided with the peak of flowering of the crop in 2016 resulted in the extensive abortion of pollinated flowers. This significantly reduced cotton production from 14.60 to 9.69 million bales. In the following year, a combination of high temperature and dry air that favored the spread of whiteflies also severely impacted cotton production in the province of Punjab [8,9]. In China, recurring cycles of drought and/or flooding have been effecting heavy yield losses to the crop [10,11].

The increasing vulnerability of cotton to emerging challenges in agriculture due to climate change is alarming, although not entirely unexpected. Like other major crops, cotton has a narrow germplasm base as a consequence of intensive selection for only a few key traits such as fiber yield and quality during domestication. This is further exacerbated by generations of industry-scale cultivation of only a few varieties with closely shared pedigrees [12]. To ensure the sustainability of cotton production amidst agro-environmental instability, novel genetic variations from diverse genetic resources that can provide wider adaptability to a myriad of environmental challenges need to be re-incorporated into the cultivated gene pool of the crop [12,13].

Landraces or traditional local varieties are a rich source of genotypic and phenotypic diversity that can be exploited to enhance the adaptability of crops to limiting environments. They are usually composed of population complexes that were not formally bred for optimum yields but were unconsciously selected for their productivity across varying seasonal environments within specific localities under traditional farming systems [14,15]. Because selection is focused not just on a single trait but on the overall fitness of the plant under a specific agro-ecological zone or eco-geographic environment, landraces are able to maintain broader genetic diversity compared to the varietal products of contemporary breeding. This make landraces an ideal germplasm to mine for adaptive traits and novel alleles for resilience to marginal environments [16].

The successful use of landraces in breeding for target trait improvement has been extensively documented in a multitude of species including cereal and leguminous crops [17,18,19,20,21,22]. In rice for example, major QTLs conferring tolerance to flooding (*Sub1 A*) and soil salinity (*Saltol*) have been identified from the landraces FR13 and Pokkali, respectively. Breeding lines derived from both landraces have been successfully used as donors in marker-assisted introgression to improve the adaptability of elite rice cultivars to flash floods and salt stress [17,18]. Additionally, genes for blast resistance such as the quantitative *pi21* allele from the landrace Sensho have been transferred by marker-assisted backcrossing to various rice cultivars. Backcross inbred lines with *pi21* exhibited resistance against a broader spectrum of blast races [19]. In peas, breeding lines derived from crosses between the elite pea cultivar Messire and five landraces (Ps624, Ps423, P665, P675 and P660) demonstrated increased resistance to the parasitic broomrape (*Orobanche crenata*) under field and controlled conditions. Under severe broomrape infestation, the breeding lines out-yielded Messire, although not when the weed pressure was low [22]. 

In cotton, the utilization of landraces in breeding efforts have yet to be exploited. This is mainly due to challenges that relate to the breeding phenology of the plant, as well as the lack of available information for the efficient identification of suitable donor parents for breeding. In this review, we discuss the potential value of landraces for target trait improvement of cultivated cotton, as well as the major challenges and current status of their utilization in breeding programs. Gene bank mining tools, i.e., Focused Identification of Germplasm Strategy and Environmental Association Analysis that have been developed to effectively screen and evaluate the genetic merit of large germplasm collections in the absence of phenotypic data are described. Both strategies capitalize on geo-reference data that are available for conserved accessions to mathematically model associations between the environment and traits related to local adaptation. Finally, we highlight the potential applications of both tools for screening cotton landraces in search of donor parents with potential values for biotic and abiotic stress tolerance breeding.

## 2. Cotton Landraces: Diversity, Traits and Challenges in Breeding Applications

Cultivated cotton is an important commodity in the global economy. It is primarily cultivated as a major source of natural fiber for textile production, as well as of oil and protein byproducts that are used as food and feeds [23]. To date, cotton is grown in approximately 29.3 million ha of land in more than 61 countries worldwide [24]. 

In the last century, concerted efforts among cotton-growing countries to collect and preserve cotton germplasm as a way of safeguarding the resiliency and productivity of the crop against possible, adverse consequences of genetic uniformity have considerably grown [25]. Repositories for cotton germplasm that includes breeding lines, exotic species, landraces, and genetic and cytogenetic stocks have been established in India, China, US, Brazil, Pakistan, Uzbekistan, France, Russia, Turkey, Greece, Mexico and Argentina [25,26]. With the number of conserved accessions estimated at 104,780, cotton ranks among the top ten crops with the largest germplasm collection worldwide along with major cereal (i.e., wheat, rice, corn, barley and sorghum) and leguminous species (i.e., beans, soybeans and peanuts) [27]. In the US, the collection, maintenance and characterization of cotton genetic resources that represent 45 species of *Gossypium* are entrusted to the US National Cotton Germplasm Collection (NCGC) [25,28]. Initially housing only cotton germplasm that were acquired from specimen collection trips, the NCGC has since expanded to include genetic resources obtained through exchanges with the national gene banks of the governments of India, China, Uzbekistan and Russia. To date, NCGC maintains approximately 10,000 accessions of cotton, with landraces comprising approximately 34.20% of the total collection [29]. 

Independent assessments of genetic diversity contained in the subsets of cotton collections across the world have highlighted the immense intra- and interspecies variation available in the landrace accessions. In particular, molecular marker profiling of a reference set composed of 1933 cultivated and wild-type accessions of *G. hirsutum* and *G. barbadense*, as well as a diverse panel of 395 *Gossypium* accessions from the NCGC indicated a greater genetic diversity in the landraces compared to the cultivated accessions of cotton [28,30]. Similar results were obtained from genotyping 416 accessions of varieties, wild relatives and landraces of cotton from the Chinese collection using an 80K SNP array [31]. Aside from its rich genetic diversity, landraces have also been reported to harbor novel and beneficial alleles that can be used to enhance the adaptability of cotton to various environmental pressures [31,32,33,34,35]. Research to find novel sources of genetic variation that can enhance the ability of cotton to germinate under cold conditions not only established the genetic divergence of landraces from cultivated accessions of *G. hirsutum* but also demonstrated their potential to germinate even under critically low temperature [32]. A separate study demonstrated the better osmotic adjustment ability of landraces under drought stress compared to cultivated cotton [33]. Given the widespread geographical distribution of cotton, it is not surprising that landraces have evolved adaptability to specific environmental stresses including those that are prevalent in current agro-environments.

Despite the dynamic properties and immense genetic potential of cotton landraces, their utilization in actual breeding programs remains limited. One of the most important factors in cotton adaptation is its photoperiod response. In the NCGC, most cotton landraces have innate photoperiod sensitivity and require short days to flower. When cultivated in the summer under temperate environments, the longer days prevent them from flowering. Without transitioning into a reproductive phase, the landraces cannot be crossed with any cultivated cotton genotype [36]. To overcome this photoperiodicity, large backcrossing efforts were initiated in the US in the late 1970s to introgress day neutrality into landraces using photoperiod-adapted genotypes as donor parents [37]. Through this program, more than 100 day-neutral conversion lines of cotton landraces have been developed and released in the US [38,39,40,41]. Phenotypic evaluations of a number of day-neutral-converted landraces showed wide variability in a range of fiber yield and quality traits, with the conversion lines generally exhibiting reduced yield performance but improved fiber quality [42,43]. Similarly, assessment of the breeding potential of a few, naturally occurring, day-neutral landraces also showed the tendency of the germplasm to transmit alleles that negatively impact yield but positively affect fiber quality [29]. Despite the transmission of negatively correlated traits, significant variations in the phenotypic performance of natural and bred day-neutral landraces indicate the presence of novel alleles in this unique set of germplasm. Such genetic variation can be exploited to enrich the genetic pool of cultivated cotton and possibly enhance the adaptability of the crop to changing environments.

Aside from the limited capacity to directly use landraces in crossing experiments with cultivated cotton, the lack of sufficient information on the potential agronomic value of the conserved accessions also contributes to its low utilization in breeding. Selection of candidate germplasm that can be used as donor parents in breeding programs is conventionally based on phenotypic evaluation of individual accessions for target traits. Given that large-scale screening of whole collections for important agronomic characteristics is both expensive and time-consuming, a significant part of germplasm collections remains un-phenotyped. Consequently, their selection and utilization in actual breeding programs becomes severely limited. With cotton landraces, phenotypic evaluation particularly for traits that are expressed at the reproductive stage is doubly challenging due to the photoperiodicity of the plants. To aid in the mining of gene banks for candidate accessions that can be used as donor parents in cotton breeding, strategies have been developed that capitalize on the ecological information associated with curated accessions instead of phenotypic data alone. The succeeding sections will focus on two of these strategies viz. the Focused Identification of Germplasm Strategy and Environmental Association Analysis.

## 3. Focused Identification of Germplasm Strategy

The Focused Identification of Germplasm Strategy or FIGS is a gene bank mining tool that was jointly developed by the International Center for Agricultural Research in the Dry Areas (ICARDA), the Vavilov Institute, the Nordic Genetic Resources Center and the Australian Winter Cereals Collection. The primary objective in developing this strategy is to provide breeders with a tool to sort through large gene bank collections for accessions having specific traits of interests in the absence of actual phenotypic or agronomic evaluation data [44]. The approach is based on the underlying principle that environments impose selective pressures that drive the adaptive evolution and consequently, the geographical distribution of plant species. Based on this premise, FIGS capitalizes on the availability of historical climatic and edaphic data (i.e., soil properties such as chemistry, texture and topography that influence organisms and processes occurring in the soil), as well as accurate geographical positioning system to mathematically model the potential relationship between an adaptive trait (i.e., disease resistance or tolerance to extreme weather conditions) and the environment that mediated the natural selection for that specific trait. The a priori information generated from the model is then used to identify (1) collection sites that were most likely to have imposed selective pressures for the evolution of adaptive characteristics in plant populations and (2) gene bank accessions from the identified collection sites with the highest probability of having the allelic variation for the traits of interest [45,46]. Figure 1 illustrates the general process of mining gene bank collections for a particular trait using FIGS. Following the identification of a trait of interest (e.g., drought tolerance) the combination of environmental parameters that will most likely select for the trait in situ are defined. Examples of these parameters include average annual rainfall/humidity, daily temperature and light properties, as well as various soil characteristics. Promising gene bank accessions from collection sites that fit the agro-climatic and soil profiles identified to potentially impose high selection pressure for drought tolerance are then selected and actually evaluated for their fitness under drought conditions. In this manner, FIGS is able to facilitate a more rapid and focused mining of large collections for adaptive traits than was previously possible using traditional tools.

Alternatively, FIGS can also benefit from limited phenotypic data that is available even for a small subset of germplasm collection [47]. In a FIGS search for insect resistance for example, the collection site for each phenotyped accession is first identified. The eco-geographic profiles associated with the collection site for each screened accession are then defined (Figure 2). Predictive models that can distinguish resistant from susceptible accessions are developed based on the eco-geographic profiles that are most likely to drive the selection for the trait in situ. The model is then applied to a larger set of un-phenotyped germplasm collection to identify the ‘best-bet’ accessions possessing insect resistance. 

Following the development, testing and refinement of the approach, FIGS has been successfully used to identify novel donors for resistance to various pests and diseases of crops including wheat and barley. A FIGS search for natural resistance to Sunn pest (*Eurygaster integriceps*) on the ICARDA wheat collection of 8376 geo-referenced accessions identified a total of 534 lines with likely resistance to the pest. Candidate accessions were selected from sites with a recorded history of Sunn pest occurrence and with the agro-climatic profile that supports high population density of the insect. Initial field screening of the 534 lines reduced the selection to 57 accessions of which eight bread wheat and one durum wheat were determined to possess resistance to the pest at the vegetative stage. These lines are currently being used as donors for resistance to Sunn pest, a major pest of wheat causing 100% yield losses in East Europe and West and Central Asia [48]. Using the same approach, donors for powdery mildew (*Blumeria graminis* (DC) Speer f.sp. *tritici*) [47,49], stem rust (*Puccinia graminis* Pers.) [50] and Russian wheat aphid (*Diuraphis noxia* Kurdj.) [51] afflicting wheat, and net blotch (*Pyrenophora teres* Drechs.) infecting barley [52] were identified from smaller subsets of germplasm collected from locations sharing similar environmental variables as the collection sites of a reference set of accessions with known resistance to the aforementioned biotic stresses. 

Aside from resistance to phytophagous pests and pathogens, a set of fava bean germplasm with traits that are related to drought adaptation has been successfully identified using the FIGS approach. Using combined geographical imaging system and agro-climatic data, a collection of 9545 accessions of fava beans were separated into two different subsets. The first subset originated from environments with low moisture content and therefore are likely to possess drought tolerance while the second subset was sourced from locations with higher moisture content. Actual evaluation of both subsets for morphological and physiological characteristics that are related to water use efficiency showed differentiation in both sets that are in agreement with the groupings obtained using FIGS. The result of the study demonstrates the effectivity of FIGS in enhancing the discovery and deployment of new genes regulating traits of interest [21]. It also underscores the presence of specific adaptabilities of landraces to local environments and therefore their suitability as a genetic resource for abiotic stress tolerance.

Although the use of FIGS has facilitated the selection of plant accessions possessing target traits, it should be noted that the tool is not without its weaknesses [53,54]. The efficiency of FIGS to mine large germplasm collections is based on the a priori knowledge of environmental pressures that drive natural selection for a specific trait [45,46]. However, there is also a possibility that target traits may have evolved not only as an adaptive response to natural selection but also in response to non-adaptive evolutionary processes such as gene flow and genetic drift. In such cases, target traits might not have strong associations with eco-geographical and eco-climatic variables. To account for the contribution of non-adaptive processes in shaping the geographical distribution of traits and enhance FIGS precision in identifying accessions with adaptive traits, the use of biogeographic variables or proxies for gene flow and genetic drift in conjunction with eco-geographical and eco-climatic profiles have been proposed. Proxies that can be considered to estimate the effects of genetic drift include habitat isolation and geographic distances between populations. For genetic drift, population size and founder effects can be used as proxies [53,54].

## 4. Environmental Association Analysis

Environmental association analysis (EAA), also known as environmental association mapping or genome-environment association, is an approach that aims to identify functional polymorphisms and genetic variants associated with local adaptation to diverse climatic and edaphic factors. This framework is under the domain of landscape genomics, which has a more general scope of identifying environmental factors that shape adaptive genetic variation [55,56,57,58]. The earliest use of landscape genomics focused on modelling the relationship between observed variations in phenotype and the environment [58,59,60]. In recent years, the development of draft and reference genomes, as well as the continuous advances in sequencing technologies improved the resolution of landscape genomics to identify allelic variations that are strongly associated with environmental variables, hence the refinement of the strategy into EAA [52,56,61]. The application of EAA in breeding is both timely and relevant amidst the backdrop of changing agro-environmental conditions due to major shifts in climate. Recent studies have shown that rising temperatures and unpredictable precipitation patterns force phenotypic plasticity in crops that allow them to adapt to changes in the environment [61,62]. These are supported by findings from multiple, separate research indicating the role of environmental variables in shaping the genetic diversity of major crops [57,63] and identifying temperature and precipitation as major contributors of observed genomic variations underlying local adaptation [58,63].

Like FIGS, EAA capitalizes on the availability of geo-referenced genetic materials, mostly a set of landraces that were sourced from stressful and diverse agro-ecological zones. Using the geographical coordinates of the collection site for each landrace, data on multiple environmental, soil and topographic parameters are collected from publicly available databases. Information on bioclimatic variables for example, are freely available and can be downloaded from the WorldClim database [64,65], whereas soil-related information are accessible through ISRIC (World Soil Information database) [66,67]. Data curation can be as detailed as including yearly, quarterly, and monthly temperature and precipitation at a resolution of 30 arc-seconds (approximately 1 km grids) [65]. 

The EAA framework follows the same underlying principle as the typical genome-wide association study (GWAS) except that phenotypic data is replaced with environmental variables [56]. In the GWAS framework, statistical analyses are used to model associations between phenotype and sequence data, with the goal of finding functional polymorphism underlying specific variations in phenotype [68]. Under the EAA framework, the environmental variables, instead of phenotypic data, are combined with sequence data [56] generated from re-sequencing, genotyping-by-sequencing, or array-based genotyping technology. Like GWAS, EAA depends on linkage disequilibrium between genetic markers that were used for genotyping and functional polymorphism to narrow down plausible candidate genes controlling local adaptation. Rellstab et al. [56] reviewed and summarized the available statistical tests that can be used for EAA, including categorical test, logistic regression, matrix correlation, general linear models, and mixed effect models. More recently, Cortes at al. [68] reviewed new mixed model frameworks and advances in the improvement in speed and power of GWAS that could be extended and applied to EAA. 

To date, the body of work on EAA is steadily growing. Applications of EAA on the world’s staple crops such as maize [57], sorghum [58], soybean [62,63] and barley [69] has revealed the complex genetic architecture underlying local adaptation in these species. In soybean for instance, several putative loci contributing to local adaptation were identified by combining genotype data for 3000 soybean landraces available at the USDA Soybean Germplasm Collection and variables such as altitude, rainfall and temperature [63]. The largest effect locus explained <5% of the total variation observed for mean precipitation in the wettest quarter. In a separate study led by the International Maize and Wheat Improvement Center, standard and geographical GWAS was used to determine the genetic drivers of local adaptation in 4471 maize landraces using latitude and altitude as parameters [57]. Latitude and altitude are negatively correlated with temperature and day length and therefore contribute in the positive selection of flowering traits that are critical for local adaptation [54,57]. Mapping of genomic regions regulating latitudinal and altitudinal adaptation identified a total of 1498 genes, nearly 40% of which overlapped with genes for flowering time [57]. 

EAA is also a powerful tool to identify genetic variants and validate known genes involved in local adaptation. Using environmental variables collected from over 1900 geo-referenced landraces of sorghum, Lasky et al. [58] genetically mapped natural variants of the previously cloned genes, i.e., *Maturity 1* and *Tannin 1* which have been associated with agro-climatic traits in sorghum. Findings showed that *Maturity 1* was most strongly associated with the minimum temperature of the coldest month. *Tannin 1* was strongly associated with a number of bio-climatic gradients, with strongest associations with mean temperature of the warmest quarter [58]. Similar studies identified other large-effect genes related to local adaptation that has been previously mapped or cloned. This includes the *Vegetative to generative transition 1* (*Vgt1*) [70,71], a cis-regulatory element involved in the regulation of flowering time in maize. *Vgt1* has been found to be evolutionary conserved across the maize–sorghum–rice lineages [70]. Navarro et al. [57] also identified *ZCN8* which encodes the maize florigen [72,73] and *ZmCCT* [74], a well-documented main-effect QTL underlying photoperiodism in maize using the landscape genomics platform. 

Lastly, applications of EAA has also demonstrated its ability to identify novel candidate genes underlying local adaptation. Bandillo et al. [63] identified a few notable abiotic-stress responsive genes related to drought and cold tolerance, including *DREB2A* [75], *AtCLB* [76], and *CAMTA* [77]. Similarly, Lasky et al. [58] identified abiotic stress responsive genes, including *OREI* and *ORS1*, which regulate drought responses in rice [78], and grain-filling and senescence in wheat [79]. SNP associations with edaphic gradient or soil condition also led to the identification of putatively important genes for local adaptation. Bandillo et al. [63] identified *RHD3* that plays an important role for root growth and development [80]. Lasky et al. [58] identified SNPs that are associated with aluminum toxicity in Arabidopsis with specific variants showing greater reduction in root growth rate in response to soils with high pH. Specifically, the SNP was found within the xyloglucan endotransglycosylase (*XET*) and *STAR1* loci, which regulates aluminum tolerance in Arabidopsis [81] and rice [82]. If these narrowed genes are taken beyond exploratory analysis and functionally validated, they could provide a better understanding of the molecular underpinnings of abiotic stresses and a useful source of variation for crop genetic improvement. 

## 5. Perspectives on the Application of FIGS and Landscape Genomics in the Focused Selection of Cotton Landraces with Target Traits of Interest

Improving baseline production and product quality remains a focal point for cotton breeding. However, requirements for the adaptability of new cotton varieties to temperature extremes, reduced precipitation and soil salinization are becoming a necessity as climate change intensifies the extent and magnitude of current and emerging agro-environmental challenges. In the Southern High Plains for example, the suboptimal temperature that bookends the short growing season has become more unpredictable in recent years, further narrowing the window for cotton planting in the region. The shorter growing season inevitably exposes the cotton plant to cold stress either during the germination or the boll maturation stage [83]. Increases in the average annual temperature have also resulted in the more frequent occurrence of drought, which has devastated cotton production in recent years [6,7]. Estimates show that a continuous rise in temperature will accelerate the depletion of important water sources like the Ogallala aquifer which provides water for approximately 20% of irrigated lands in the US including the Texas High Plains. Because 80% of water drawn from the aquifer is used for agriculture, depletion of this water resource is predicted to lead to a downturn not only in cotton productivity but in agricultural output as a whole [84,85]. Compounding these abiotic challenges are risks of disease and pest outbreaks that can potentially damage the crop in the years to come. In 2017, Fusarium wilt caused by the fungal pathogen, *Fusarium oxysporum* f. sp. *vasinfectum* Race 4 (FOV4) was identified in several fields in El Paso and Hudspeth counties in Texas. This presents a significant threat to the major cotton producing region in the US. Unlike most races of *Fusarium*, FOV4 survives in soils with neutral to alkaline pH and does not require the presence of root-knot nematodes to infect cotton plants. Because other plants are able to host FOV4 without getting infected, the pathogen can no longer be eliminated once it is introduced in the field. To date, cotton cultivars with resistance to FOV4 are unavailable, and no form of chemical or cultural control have been effective in managing the disease [86].

Amidst the burgeoning threats to cotton production, there is a growing interest in identifying new sources of genetic variation that can be used to enhance the adaptability of the crop to marginal environments. Panels of mutant lines and obsolete cotton varieties have been evaluated for their physiological responses to a multitude of environmental challenges including cold and drought stress but not the landraces [32,34,35].

Breeding for complex traits requires the effective identification of potential donor parents that can provide significant genetic gains in the long-term. Landraces possess both genetic heterogeneity and local adaptation that make them an ideal germplasm source for breeding climate-resilient cultivars. Although they have been largely replaced by modern, high-yielding varieties, a substantial portion of genetic diversity from crop landraces has been maintained in germplasm banks across the world. 

FIGS and EAA are gene bank mining tools that have been developed to circumvent the most significant challenge in utilizing germplasm collections—the phenotyping bottleneck resulting in the lack of agronomic data for relevant breeding traits. Without information on the potential breeding value of conserved germplasm accessions like landraces, selection of potential donor parents for breeding would be a futile exercise of finding needles in a haystack. FIGS and EAA use detailed information about the geographical origin of conserved accessions, as well as the historical climatic and edaphic data associated with each collection site instead of using phenotypic data as a basis for selection. These information provide the basis to model environments where adaptive traits such as tolerance to drought, salinity and heat stress are most likely to have evolved. The main difference between the two strategies is that EAA combines genomic data with landscape variables to identify genetic variants that are strongly associated with specific environmental conditions in a geo-referenced population. This not only provides an opportunity to understand the molecular basis of adaptive responses in landraces but also to clone and functionally validate causal genes/QTLs regulating the adaptive responses prior to their utilization in breeding programs. 

The NCGC of USDA alone has curated approximately 2500 accessions of cotton landraces from diverse agro-ecologies in North and South America, Africa, Asia, Australia and Europe. Agronomic evaluation for all the maintained accessions is still lacking but core passport data for each line is available and are publicly accessible from the US National Plant Germplasm System (NGPS) database [87]. These data include detailed information on the geographical origin, taxonomy, as well as the general morphology, phenology (i.e., maturity and photoperiodic rating) and productivity of landraces. In some cases, data on the resistance of conserved accessions to plant pathogens (e.g., Fusarium wilt) that commonly afflict cultivated cotton varieties are also provided. The availability of geo-references for each conserved accession satisfies the key requirement in the use of the FIGS or EAA framework to screen landrace collections for potential donors for cotton improvement. 

For FIGS, selection of suitable cotton landraces can be initiated by filtering the collection to include only accessions with available data on longitudinal and latitudinal coordinates for collection sites. Single accessions from each set of coordinates may be selected to avoid duplicates in the panel that will be used for the search. Environmental parameters associated with the coordinates for each collection are available through WorldClim [64,65] and ISRIC [66,67]. These databases are reliable sources of detailed information on temperature, precipitation, altitude, and soil pH, bulk density, cation exchange capacity and organic and inorganic content in each collection site. Any combination of these agro-climatic parameters can be applied as filters for the determination of subsets of materials with the highest probability of possessing the trait of interest. In using eco-geographical and eco-climatic data as proxies for actual phenotyping, the costs associated with large-scale screening of cotton landrace collections for traits of interest are significantly offset. 

To increase the resolution of germplasm searches, genomic scans of the filtered subset of landraces can be performed following the framework of EAA. High density SNP chips such as the CottonSNP63K [88] and CottonSNP80K [89] are already available for cotton. Both arrays were developed using the Illumina Infinium technology based on discovery sets that represents diverse cultivated and wild cotton germplasm. Applications of the SNP arrays in GWAS analysis have successfully identified major loci regulating fiber quality and yield [31], as well as drought tolerance in populations derived from crosses with cultivated cotton varieties and landraces [31,34]. Findings from these results indicate the suitability of both SNP arrays in EAA application. Genotyping services using the CottonSNP63K are available through the Texas A&M Institute for Genomic Sciences and Society. Under the framework of landscape genomics, hierarchical population structure, linkage disequilibrium and spatial ancestry analysis; partitioning of genomic variations due to climatic variables and geographic distances; identification of targets of selection using fixation indices; environmental association analysis; haplotype analysis and candidate gene annotation and enrichment can be carried out to identify best-bet accessions and determine loci under selective sweeps for a particular trait. In addition, the landscape genomic approach could be used for characterizing and predicting genotype-by-environment interaction because of the relevance of environmental variables to abiotic stresses [56].

Germplasm subsets that have been identified to possess fitness advantage under stress (e.g., drought, heat and salinity) can then be screened and validated in common garden experiments. Most cotton landraces are photoperiod sensitive [36], making complementary validation experiments in the field challenging. To circumvent this limitation, multi-environment and multi-year screening of selected materials can be carried out via partnerships with other universities and research institutions that can provide suitable growing conditions for field evaluations. Combining resources and expertise can significantly minimize the time and cost involved in field evaluations. Additionally, phenotype data obtained under natural field conditions for subsets of landraces can be made available to the cotton research community through the establishment of public data repositories. This will maximize the impact of collaborative phenotyping efforts towards the widespread utilization of landraces in breeding. 

At the minimum, the application of FIGS to mine landrace collections of cotton will allow the strategic and more focused identification of a subset of landraces that are most likely to have the target traits of interest. Combined with genomics scans under the framework of EAA, the search resolution increases, resulting in the potential identification of candidate loci or genes underlying local adaptation to specific climatic or edaphic conditions. To improve the adaptability of cultivated varieties, the selected germplasm accessions with validated adaptive loci can be directly used in breeding programs as donor parents or for germplasm enhancement strategy. Genetic information on the validated, functional SNP variant(s) regulating the fitness advantage of a landrace to specific environments can also be used for gene-editing. Lastly, the genotype data generated through EAA can be also be used in combination with GWAS for highly complex quantitative traits using the available phenotypic data in NGPS for the extracted cotton panel.

## Figures and Tables

**Figure 1 plants-10-01300-f001:**
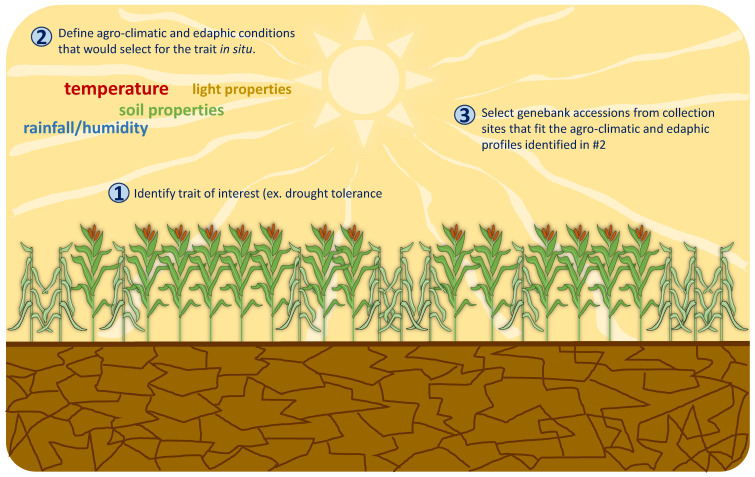
Basic FIGS search for germplasm having trait of interest. Following the identification of a trait of interest, i.e., drought tolerance, the agro-climatic and edaphic conditions under which plants are most likely to develop the trait of interest are profiled. Gene bank collections are then mined for accessions that were collected from locations with environmental profiles that fit the conditions identified to select for the trait in situ. The selected subset of best-bet accessions will be screened for drought tolerance.

**Figure 2 plants-10-01300-f002:**
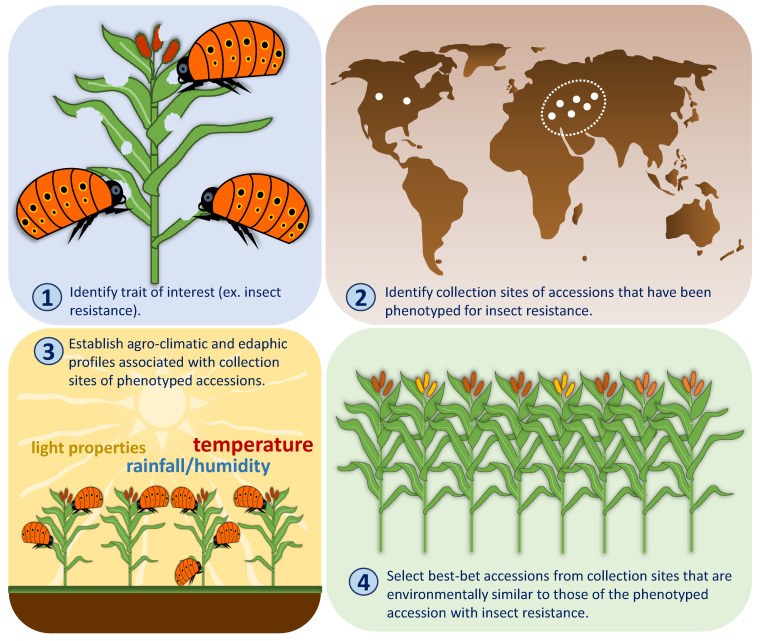
Using limited phenotype data for a FIGS search. In the presence of evaluation data for a limited number of accessions, the eco-geographical profile of the sites where the phenotyped accessions were collected can be used as a basis to identify other accessions that are most likely to possess the target trait.

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
