# Peer review of "Finding Needles in a Haystack: Using Geo-References to Enhance the Selection and Utilization of Landraces in Breeding for Climate-Resilient Cultivars of Upland Cotton (*Gossypium hirsutum* L.)"

_plants, 2021, doi:10.3390/plants10071300_

Round 1
Reviewer 1 Report
It’s a simple review, albeight on a relevant and interesting subject.
Line 120 G hirsutum and G barbadense (italic)
129 G hirsutum (italic)
137 Which collections you refer when you say “most of them” are photoperiodicaly sensitive?
230 to 233 scientific names in Itallic letters
287 new paragraph
300 which climate variables are under study?
305 please, if possible, explain why longitude and latitude were considered more usefull than particular traits, in this particular case. Do longitude and latitude where cultivars are developed strongly determine where they should be planted? Was it evaluated with any particular maize kind? Is yield a main final concern in this analysis?
308 This statement could be a conclusion for the last paragraph “Aside from identifying genomic regions under selection pressure from specific environment. Starting this paragraph with “EAA is also a powerful tool to determine whether known genes regulating abiotic stress responses are likely to be involved in local adaptation.”
316 may you briefly describe VGT1?
318 The genes ZCN8 and ZmCCT were known before?
322 Bandillo et al studied soybean... in what sense DREB is it a newly identified gene?
Reviewer 2 Report
Referee report
for manuscript entitled “Finding needles in a haystack: Utilizing geo-references to enhance 2 the selection and utilization of landraces in breeding for abiotic 3 stress tolerance in upland cotton (Gossypium hirsutum L.)”
Current manuscript describes two screening approaches for the faster evaluation of landraces with probable advantageous loci and the further implication in breeding. The first two sections are well written and composed. In the next sections, text suffers from repeated information without more detailed explanations regarding technical issues of the proposed technologies. Technically, the verbs of many sentences are placed after the additional information which complicates text comprehension. The manuscript could be greatly improved by rearrangement of the text to more focused explanations, removing redundant information, addition of details about species of interest and information about landraces, by the addition of the clearly stated summary or conclusions. There is also a lack of the discussion about drawbacks of the highlighted applications, for example, the state of gene annotation for the species of interest, impact of species genome features, current information content available. My main recommendation for the authors is to review text in the last three sections of the manuscript and focus more on review of the literature and previous findings, with less emphasis on arguing the significance of further study. Therefore, main revision is recommended. Following indications display mentioned issues.
lines 107-108- it is possible to combine two paragraphs into one.
172-176- very long complex sentence, please split it into several or reword.
200-207- assemble the sentence, which is currently separated by the figure.
213-214- please insert reference.
216- sentence is not finished and I was unable to find the continuation.
198-212- unclear description. I guess, at the initial stage, the model is trained on the accessions with known phenotype possessing the resistance, but then it could not been called as betting (in line 200). However, later algorithm is applied on the un-phenotyped landraces, where real ‘best-bet’ accessions are selected for further screening for the resistance (not yet possessing resistance as it is indicated in line 212).
Figure 1 is not informative, it is not the general overview of the process as stands in the caption, but representation of a particular case with drought tolerance selection. I suggest to improve this figure to include general scheme, or change the description of the figure to display applications in selection against abiotic stressors.
Line 230-236. “focused identification of donors….. were identified…”. Readers expect the verb in a sentence to be near the subject of that sentence. In this manuscript many sentences are with inserted long text that describes the subject between the subject and verb. Furthermore, at the beginning of particular sentence it stands, that “ Using the same approach…”, but at the end of the same sentence is inserted an additional explanation of this approach, which was already explained in details previously several times.
Line 251. identification of causal genes- very rough formulation, as usually mutations or sequence variation in the genes (alleles) are identified and also other changes could impact gene performance.
Line 262-264- this sentence could be raised at the beginning of the section as several synonyms of the method are given which could be more useful in the introduction of the approach.
Line 276- not clear, why data was downloaded.
Line 277-279 it would be useful to advert the availability of such data for the cotton since this review is about this species.
Line 284- “ polymorphisms regulating a specific variation in phenotype”. Polymorphisms per se are not regulating, but rather impacting phenotype, in the result of complex interactions.
Lines 287-288 are repetition of the lines 282-283.
Lines 304-305- verb is at the end.
Line 306-307- please review cited information, check the usage of terms- genes / polymorphisms.
Line 312-335- section needs improvement. It is not clear without the context of cited publication what is Maturity 1 and Tannin 1. Please use unite gene names in italic. Please explain in this section mentioned gene/loci names or function for the review of the information in this publication. Please use italic font for names of species in Latin.
368-369- awkward wording.
373-382- repeated information.
389- through and through?
390-395- unclear the meaning of the sentence. All those applications are aimed for the identification of the loci under the selective pressure, but best-bet accession identification is only the first step of landscape genomic.
Line 401-404 – This part resembles the project proposal, not a review of the literature- please reword or add citations.
Line 414- two verbs “be”.
Reviewer 3 Report
Authors reviewed important findings about the selection and utilization of landraces in breeding for abiotic stress tolerance in cotton by using geo-references. The whole paper was well written and arranged well too. The shortage is that less recent studies about breeding for abiotic stress tolerance were mentioned in this review, which limited the importance of this review. I suggest authors add more recent studies about breeding for abiotic stress tolerance in different plant species by using landraces to enrich this review.Author Response
Please see the attachment.

Reviewer 4 Report
This manuscript reviews two approaches, "Focused Identification of Germplasm Strategy" (FIGS) and "Environmental Association Analysis or Mapping" (EAA) to try to set a research road to deal with the homogeneity of cultivated cotton. The review is well written and addresses most approaches to uncover local adaptation in cotton landraces helping in breeding studies. This is supported by agronomic literature on FIGS and molecular ecology literature in local adaptation. I have the following comments:
- Lines 91-93, while including pest resistance and biotics stress in their analyses, there have been works which point to the limitations of the approach for breeding for pest resistance (i.e. Stenberg and Ortiz, Curr. Op. in Insect Science (2021). I suggest that a section on limitations should be included.
- Lines 124 and 385, I wonder if the number of SNPs that can be detected influence the usefulness of this approach. In particular will an 80,000 SNPs or a 63,000 panels be enough in cotton?. This should be discussed
- Lines 156-158, this problem seems insurmountable, how could we deal with it?
- Lines 213-236, it would probably be important to list the cotton pests with which a cotton breeder has to deal with instead of referencing other crops
- Lines 237-249, which biotic stress should we breed for in cotton?, see also comment 8
- Line 311, could you suggest in consequence of comments 4 and 5 a teptative proof of concept in cotton?
- Line 389, delete "through the"
- Lines 336-416, this section is too general and has to be included and detailed in the rest of the review since cotton is the study subject. For example, the reader would like to know, which abiotic stress and pests are relevant in cotton breeding to use the approaches proposed
Reviewer 5 Report
Shim et al. review two potentially useful strategies for improving the genetic diversity and resistance to adverse environmental conditions of domesticated cotton through utilization of landraces. I found the review very well written with admirably clarity given the topic. There appears to be a strong case to incorporate these approaches sooner rather than later. There are some minor suggestions below to consider that may improve the context of a few areas presented.
- I thought additional commentary on the perspective (or opportunity cost) of cotton breeders (or interested parties in the supply chain) may be beneficial to the reader. For example, consideration of the time-frame to identify advantageous traits (by FIGS, EAA) and then incorporation into a breeding program may be informative. Additionally, since fiber is still the driving force and as stated in line 146 there was a variation in yield/quality of landraces’s fiber, maybe a few comments on the opportunity costs of these approaches would provide additional content.
- Line 125 sentence implies many such examples. Two are provided after this sentence. But if there are additional examples maybe add a few references since it is an important statement within this review?
- Line 180. Given the term edaphic is somewhat specialized and used throughout I would suggest defining it at first use.
- Two additional references to consider:
- Johan A Stenberg, Rodomiro Ortiz. Focused Identification of Germplasm Strategy (FIGS): polishing a rough diamond. Current Opinion in Insect Science, Volume 45, 2021, Pages 1-6, ISSN 2214-5745, https://doi.org/10.1016/j.cois.2020.11.001. (https://www.sciencedirect.com/science/article/pii/S2214574520301450)
- Vikas, VK, Kumar, S, Archak, S, et al. Screening of 19,460 genotypes of wheat species for resistance to powdery mildew and identification of potential candidates using focused identification of germplasm strategy (FIGS). Crop Science. 2020; 60: 2857– 2866. https://doi.org/10.1002/csc2.20196
Round 2
Reviewer 2 Report
The manuscript has been supplemented, suggestions are taken into account and errors have been corrected, therefore I have no any objections to the publication of this manuscript.